



# Modelling the artificial forest (*Robinia pseudoacacia* L.) root-soil water interactions in the Loess Plateau, China

Hongyu Li[1,2], Yi Luo[1,2,3], Lin Sun[1], Xiangdong Li[4], Changkun Ma[5], Xiaolei Wang[1,2], Ting Jiang[1,2], Haoyang Zhu[1,2]

[1] Key Laboratory of Ecosystem Network Observation and Modeling, Institute of Geographic Sciences and Natural Resources Research, Chinese Academy of Sciences, Beijing, 100101, China
[2] University of Chinese Academy of Sciences, Beijing, 100049, China
[3] Research Centre for Ecology and Environment of Central Asia, Chinese Academy of Sciences, Urumqi, 830011, China
[4] Guangdong Key Laboratory of Integrated Agro-environmental Pollution Control and Management, Guangdong Institute of Eco-environmental Science & Technology, Guangzhou, 510650, China
[5] State Key Laboratory of Eco-hydraulics in Northwest Arid Region, Xi'an University of Technology, Xi'an, 710048, China

*Correspondence to*: Yi Luo (luoyi@igsnrr.ac.cn)

**Abstract.** Plant root–soil water interactions are fundamental to vegetation–water relationships. Soil water availability and distribution impact the temporal–spatial dynamics of roots and vice versa. In the Loess Plateau (LP) of China, where semiarid and arid climates prevail and deep loess soil is dominant, drying soil layers (DSLs) have been extensively reported in artificial forest land; however, the underlying mechanism remains unclear. This study proposes a root growth model that simulates both the dynamic rooting depth and fine root distribution, coupled with soil water, based on cost–benefit optimisation. Evaluation of field data at an artificial forest site of black locust (*Robinia pseudoacacia* L.) in the southern LP positively proves its performance. Further, a long-term simulation was performed to address the DSL issues, which were forced by a 50-year climatic data series, under variations in precipitation. The results demonstrate that incorporating the dynamic rooting depth into the currently available root growth models is necessary for reproducing the drying soil processes. The top 2.0 m is the most active zone of infiltration and root water uptake, and below which the fractions of fine roots and uptake are small but cause a persistently negative water balance and consequent DSLs. The upper boundary of the DSLs fluctuates strongly with infiltration events, while the lower boundary extends successively owing to the interception of most infiltration by the top 2.0 m layer. Coupling the root–water interactions helps to reveal the intrinsic properties of DSLs, with the persistent extension of its thickness and rare opportunities for recovery from the drying state. This study may have negative implications for the implementation of artificial afforestation in this semiarid region, as well as in other regions of similar climate and soils.

**Key words:** Root–soil water interaction; Root growth model; Cost–benefit optimisation; Drying soil layer; Loess Plateau



# 1 Introduction

Plant roots are a significant pathway of the soil–plant–atmosphere continuum (SPAC), which connects the aboveground parts of the plant and the soil environment (Feddes et al., 2001; Mencuccini et al., 2019) by extracting water from the soil to meet the evaporation demand of the canopy. This soil water uptake process is regulated by root profile properties, which are

highly dynamic in response to variable soil water conditions (Schenk and Jackson, 2002; Fan et al., 2017). In particular, forest root structures are rather complex (e.g., have woody coarse roots for anchoring and non-woody fine roots for absorption), which enable diverse water exploration strategies for adaptation to changing environments (Mulia and Dupraz, 2006; Ivanov et al., 2012; Sivandran and Bras, 2013; Brunner et al., 2015). For example, in order to increase water uptake, forests tend to grow more roots in the wetter soil layers (i.e., root hydrotropism) or develop deep roots to extract deeper

water resources (including deep soil water and groundwater) (Maeght et al., 2013; Bardgett et al., 2014; Phillips et al., 2016). The investigation also indicated that forest stands develop complicated morphological distribution of roots and diverse root water uptake strategies to adapt to the diverse soil water status (Germon et al., 2020; Knighton et al., 2020). Therefore, plant root–soil water interaction is a key issue for understanding the forest–water relationship, which is inevitably an important part of ecohydrological models, fundamentally for plant water uptake (Smithwick et al., 2014).

Plant water uptake is usually taken as a sink term in water movement equations (Feddes et al., 2001; Clark et al., 2015). The sink term is expressed as a function of the morphological and hydrological traits of the roots and soils. Morphologically, the root profile contains two primary features, rooting depth and vertical distribution (Warren et al., 2015), which are commonly included in most current root uptake models. These features are usually considered static in most of the available hydrologic and terrestrial biospheric models (Luo et al., 2003; Warren et al., 2015). The maximum rooting depths are

generally assumed to be static values which may differ from the plant functional types (Ostle et al., 2009). Meanwhile, the vertical root distribution is represented as an empirical function of root length density to soil depth over the root domain (Jackson et al., 1996; Zuo et al., 2004; Sivandran and Bras, 2012), which describes the morphological features of roots statically. These simplifications of the root features allow ease in practical applications to simulate the root uptake process. However, it is increasingly recognised that efforts should be made to account for root dynamics, especially when the

coupling effects between plant growth and water availability are considered (Warren et al., 2015).

The dynamic roots indicate that the hydrologic or terrestrial biospheric models simulate growing roots under changing environmental conditions, for example, soil water status. This is further incorporated into the root water uptake models. In these process models, the dynamic root profiles may be accounted for by either the changing rooting depth (Gayler et al., 2014; Hashemian et al., 2015; Christina et al., 2017; Liu et al., 2020) or root density distribution (Schymanski et al., 2008;

Wang et al., 2016; Wang et al., 2018a; Drewniak, 2019; Niu et al., 2020), which is not sufficient to describe the dynamic root adaptation to changing environmental conditions (Rudd et al., 2014). The rooting depth, root density distribution, soil water quantity, and its spatial distribution are interrelated, and their coupling should be reflected in the root water uptake modelling (Warren et al., 2015).



To meet the water requirement, plants tend to develop more roots in water-rich zones (Germon et al., 2020). Within the root system, soil water is conveyed up to the aboveground parts through fine and coarse roots, the former for water uptake (commonly considered as root with a diameter smaller than 2 mm) and the latter for water transport. Fine roots are developed on coarse roots, together constituting a hydraulic architecture, creating structural relationships for water transport (Smithwick et al., 2014; Chen et al., 2019). According to the pipe model theory (Lehnebach et al., 2018), for denser fine roots to uptake more soil water, coarser roots are required to maintain the hydraulic transport capacity, especially for deeper extension. Balancing the cost of biomass allocation to coarse or fine roots and the benefit of the water taken up is a great ability of plants which live under water-stressed conditions (Guswa, 2008). Mathematical optimisation methods have been widely implemented in previous studies to estimate the optimal root profiles (Kleidon and Heimann, 1998; Collins and Bras, 2007; Guswa, 2008; Schymanski et al., 2008 and 2009; Yang et al., 2016). Nonetheless, optimisation efforts for root dynamic processes remain limited. Thus, the optimisation method potentially beneficial for simultaneously estimating the dynamics of rooting depth and fine root distribution.

Black locust (*Robinia pseudoacacia* L.) trees have been widely planted in the Loess Plateau of north-western China to prevent serious soil erosion (Jia et al., 2017a). The Loess Plateau is in a semi-humid and semiarid zone. Its deep loess soil has a significant capacity for water storage. The high water demand drives the planted trees to grow roots to seek water in deeper soil layers, creating a well-developed root system with a strong water absorption ability (Vitkova et al., 2017). After years of growth, in situ investigations have indicated that deep soil desiccation (drying soil layers) has developed extensively in these forestlands (Deng et al., 2016; Jia et al., 2017b; Liang et al., 2018; Wu et al., 2021), and premature and mortality of the planted black locusts occurred due to soil water stress (Zhang et al., 2020). However, how the black locust roots and soil water interact has not been addressed in previous modelling studies (Liu and Shao, 2015; Tian et al., 2017; Turkeltaub et al., 2018; Li et al., 2019a; Bai et al., 2020). It is presumed that root–water coupling may play a key role in the ecohydrological processes in the black locust forestland.

Therefore, this study aimed to (1) develop a root growth model based on the cost–benefit theory, which can simultaneously adjust root distribution and rooting depth in the root water uptake model, and (2) reveal the black locust root–soil water interactions over the soil profile based on a long-term simulation to address the drying soil layer issues in this region.



## 2 Materials and methods

### 2.1 Model development

#### 2.1.1 General description to the ecohydrological model

Fundamentally, this ecohydrological model is an integration of different components from the Soil and Water Assessment
Tool (SWAT) (Neitsch et al., 2011; Arnold et al., 2012) and Community Land Model version 4.5 (CLM4.5) (Oleson et al., 2013) (Fig 1). On this basis, root growth and root uptake modules were modified. Furthermore, an optimisation approach was introduced to simulate the coupled effects of root growth and soil water dynamics.

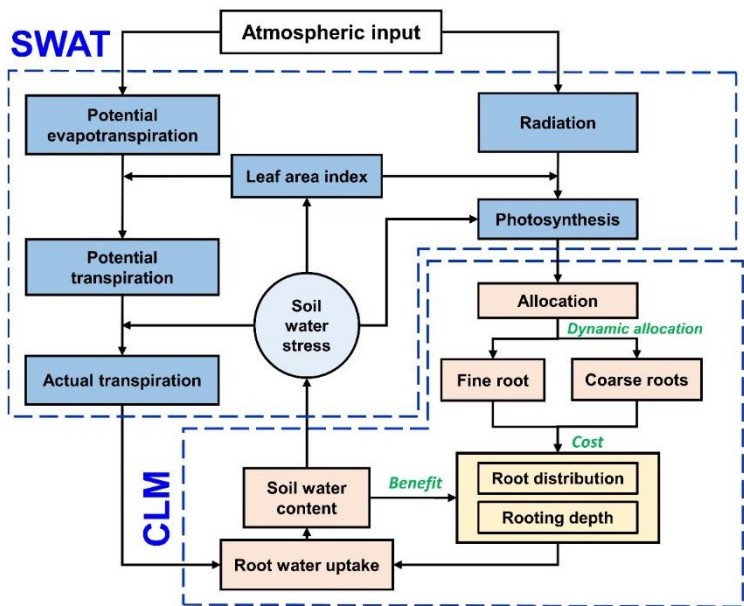

**Figure 1: The model structure integrated from SWAT (blue boxes) and CLM (light orange boxes) components. The root growth**
**module is highlighted in yellow, and the descriptions of the dynamic fine root distribution and rooting depth approach are**
**illustrated as green text.**

Hydrologically, surface process modules, for example, simulating evaporation, transpiration, canopy interception, and runoff, follow the SWAT model which is detailed in its theoretical document (Neitsch et al., 2011). The subsurface
hydrological modules, primarily the soil water movement, adopted the 1-D Richards' equation. It is solved numerically following the finite difference scheme used in CLM4.5 (Oleson et al., 2013).

Biologically, modules for the aboveground parts, for example, plant phenology, leaf area index (LAI) development, and biomass accumulation, are adopted from SWAT (Neitsch et al., 2011) and in CLM 4.5 (Oleson et al., 2013). The module for the belowground part, the root growth model, simulates root growth following CLM 4.5 which adopts the approaches from



process simulation directly from the implementation in the Biome-BGC (Biome BioGeochemical Cycles) model (Thornton et al., 2002). See the **Supplementary File** for more information on the detailed model descriptions. In CLM 4.5, the roots are categorised into coarse and fine. The rooting depth is presented as a static value, and the fine root distribution follows a static shape function which defines the root density over the soil profile.

The root–water interaction is integrated into the root uptake model, which uses the rooting depth and fine root distribution as input and acts as a sink term in the Richards' equation. Soil water imposes water stress on the root growth. The cost of biomass invested in coarse and fine roots and the benefit of water uptake were optimised through a cost–benefit function.

The following sections will describe three root growth approaches in a stepwise manner: (1) the static root distribution approach implemented in CLM4.5, which assumes a static rooting depth of the coarse roots and distribution of fine roots (Oleson et al., 2013); (2) the approach that assumes a dynamic distribution of fine roots but a static rooting depth (Drewniak, 2019); and (3) the approach proposed in this study that assumes a dynamic distribution of fine roots and rooting depth of coarse roots. These three root growth modelling approaches are incorporated into the eco-hydrological model mentioned above for a comparison of their performances.

### 2.1.2 Static distributions of coarse and fine roots

In this approach, a static exponential function expresses the root fractions in different soil layers (Zeng, 2001):

$$
fr_i = \begin{cases} 0.5[\exp(-r_a z_{h,i-1}) + \exp(-r_b z_{h,i-1}) - & 1 \leq i < n \\ \quad \exp(-r_a z_{h,i}) - \exp(-r_b z_{h,i})] & \\ 0.5[\exp(-r_a z_{h,i-1}) + \exp(-r_b z_{h,i-1})] & i = n \end{cases} \tag{1}
$$

where $i$ is the sequential number of soil layers, $n$ is the total number of soil layers in the rooting zone, $z_{h,i}$ is the depth of soil layer $i$, $r_a$ and $r_b$ are two shape parameters. The shape parameters can be obtained by fitting to the observations for different plant types, set as 6.0 and 2.0 for deciduous broadleaves in this study, respectively (Oleson et al., 2013).

### 2.1.3 Dynamic distribution of fine roots

This approach assumes that newly assimilated biomass to the belowground parts is allocated to develop fine and coarse roots with a static ratio. In general, there is a linear relationship between the carbon mass and biomass (Niu et al., 2020). Therefore, the term "carbon" will be used in the following text to refer to the biomass of different plant components. Within the present rooting depth (static over the simulation period), the fine roots are distributed over the soil depth according to the soil water content. In each soil layer $i$, the fine root carbon increment is updated at each time step, as follows:

$$
FR_{i,t} = FR_{i,t-1} + \Delta FR_{i,t} \tag{2}
$$





where $FR_{i,t-1}$ (g m$^{-2}$ d$^{-1}$) is the fine root carbon of soil layer $i$ during the previous time step, $\Delta FR_{i,t}$ (g m$^{-2}$ d$^{-1}$) is the newly allocated carbon to fine roots of soil layer $i$ at time $t$, which is modified by soil water content:

$$\Delta FR_{i,t} = \Delta FR \cdot \frac{THK_i REW_i}{\sum_{i=1}^{n}(THK_i REW_i)} \tag{3}$$

where $\Delta FR$ is the newly assimilated carbon allocated to total fine roots (g m$^{-2}$ d$^{-1}$), $n$ is the total number of soil layers in the rooting zone, $Thk_i$ is the soil layer thickness of soil layer $i$ (cm), $REW_i$ is the relatively effective soil water content (i.e., soil water availability) given by:

$$REW_i = \frac{\theta - \theta_{wp}}{\theta_{fc} - \theta_{wp}} \tag{4}$$

where $\theta$ is the soil water content (cm$^3$ cm$^{-3}$), and the subscripts $wp$ and $fc$ indicate the soil water content at the wilting

point and field capacity, respectively.

The fine root fraction in each soil layer $i$ is then calculated:

$$fr_i = \frac{FR_{i,t}}{\sum_{i=1}^{n} FR_{i,t}} \tag{5}$$

### 2.1.4 Dynamic distributions of coarse and fine roots

This approach assumes that both the rooting depth of coarse roots and the distribution of fine roots change with soil water. In

formulating the growth of the coarse and fine roots, the newly allocated biomass/carbon for the belowground part is optimally allocated to the roots based on a cost–benefit function that will be described in detail.

### (1)  Carbon allocation between the coarse and fine roots

The cost is defined as the amount of carbon invested to grow coarse/fine roots. In constructing the coarse and fine root systems, the pipe model theory (PMT, Shinozaki, 1964) was adopted. Lehnebach et al. (2018) summarised "the essence of

the PMT concept" as "*a unit amount of leaves is provided with a pipe whose thickness or cross-sectional area is constant. The pipe serves both as the vascular passage and as the mechanical support and runs from the leaves to the stem through all intervening strata.*" The relationship between the leaf and stem can be established quantitatively based on the PMT and can be extended to the belowground parts of the plants (Chen et al., 2019). For roots, the relationships have been established in analogue form and validated against some databases (Carlson and Harrington, 1987; Richardson and Dohna, 2003).

Delineating the soil profile into adjacent soil layers which correspond to the numerical solution of the Richards' equation, the equations for modelling the root growth are also written regarding the discrete soil layers (Fig 2c).


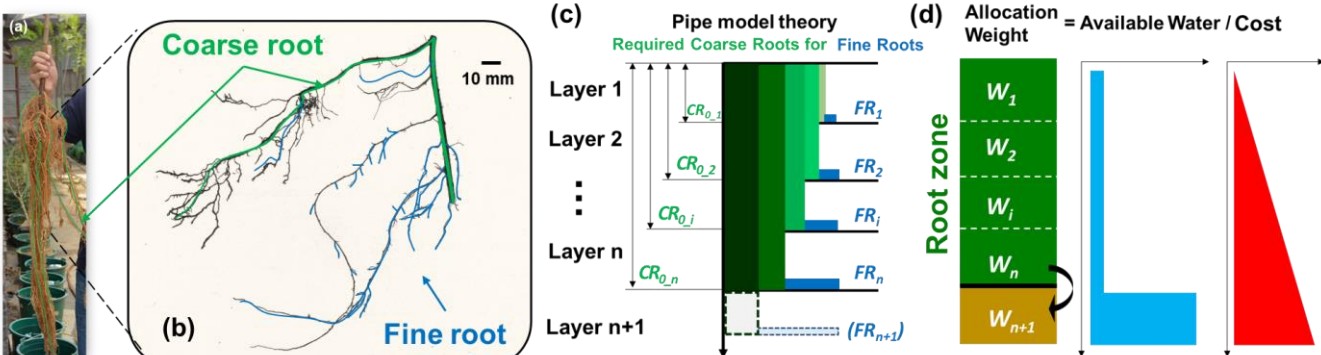

**Figure 2: (a) Black locust root system obtained from an experimental pot; (b) scanned image of one root segment consisting of coarse and fine roots; (c) schematic description of pipe model theory adopted for coarse and fine roots; (d) weighting factors for biomass allocation and conceptualized benefit (available soil water for uptake) and cost (biomass investment)**

The biomass amount for coarse and fine roots should be maintained as

$$CR_{0\_i} = \rho \cdot Z_i \cdot k_A \cdot FR_i \tag{6}$$

where $CR_{0\_i}$ is the equivalent carbon for coarse roots in the thickness between the ground surface and bottom of soil layer $i$, $FR_i$ is the carbon for fine roots within the soil layer $i$, $\rho$ is the mass density of coarse roots (g/cm³), $Z_i$ is the depth from the surface to the bottom of soil layer $i$ (cm), *and $k_A$ is a constant which can be determined from field observations (see Section 2.3.2).

For an increment of carbon for fine roots within soil layer $i$, a corresponding increment of carbon for coarse roots is needed and can be derived from Eqn. (6):

$$\Delta CR_{0\_i} = \rho \cdot Z_i \cdot k_A \cdot \Delta FR_i \tag{7}$$

Notably, there is no increment of carbon for coarse roots if the available coarse roots are sufficient to support the new fine roots, as discussed later.

The increments of carbon for fine and coarse roots were then summed over the root profile:

$$\Delta CR = \sum_1^n \Delta CR_{0\_i} \tag{8}$$

$$\Delta FR = \sum_1^n \Delta FR_i \tag{9}$$

The sum of the newly allocated carbon for fine and coarse roots is equal to that allocated to the belowground part $\Delta TR$:

$$\Delta FR + \Delta CR = \Delta TR \tag{10}$$

Two ratios are defined for fine and coarse roots, respectively as:

$$K_{FR} = \frac{\Delta FR}{\Delta CR + \Delta FR}, \quad K_{CR} = \frac{\Delta CR}{\Delta CR + \Delta FR} \tag{11}$$

**(2) Spatial distribution of new fine and coarse roots**

Potentially, more fine roots develop in wetter soil zones to gain more water and reduce water stress to the least extent. It is fundamentally recognised that penetration into deeper soil requires biomass for the fine roots, as well as the corresponding





coarse roots; the potential benefit can be more water uptake from the deeper soil. The basic principle for the distribution of the roots, either fine only or both coarse and fine, is that an optimal distribution of new roots helps to gain as much water as possible (Fig 2d).

Thus, the distribution of fine roots is influenced by a benefit-to-cost ratio, defined as:

$$W_i = \frac{REW_i}{CFR_i} \tag{12}$$

where $W_i$ is the benefit to cost ratio in soil layer $i$, and the benefit is presented by $REW_i$, as defined previously; $CFR_i$ is defined as the marginal carbon cost of fine roots:

$$CFR_i = \frac{\partial(\Delta CR_{0\_i} + \Delta FR_i)}{\partial \Delta FR_i} = \rho \cdot Z_i \cdot k_A + 1 \tag{13}$$

Combining Eq. (13) and Eq. (12):

$$W_i = \frac{REW_i}{\rho \cdot Z_i \cdot k_A + 1} \tag{14}$$

Replacing $REW_i$ with $W_i$ in Eq. (3), the fraction of new fine roots in the soil layer $i$ becomes

$$\Delta FR_i = \Delta FR \cdot \frac{THK_i W_i}{\sum_{i=1}^{m}(THK_i W_i)} \tag{15}$$

Updating the fine root fractions using Eq. (7), the demand for coarse roots that can meet the hydraulic transport demand
for the fine roots within and below soil layer $i$ is

$$CRP_i = \sum_{j=i}^{n} CR_{0\_i,NEW} = \sum_{j=i}^{n} \rho \cdot Z_i \cdot k_A \cdot (FR_i + \Delta FR_i) \tag{16}$$

Comparing the demand for and current storage of coarse roots, the coefficient of the coarse root carbon increment regarding the soil layer $i$ is calculated as

$$\beta_i = \frac{\max(CRP_i - CR_i, 0)}{\sum_{i=0}^{n} \max(CRP_i - CR_i, 0)} \tag{17}$$

According to Eq. (11), the increment of carbon for the coarse roots with respect to each soil layer is then calculated by:

$$\Delta CR_i = \Delta TR \cdot K_{CR} \cdot \beta_{Ci} \tag{18}$$

**(3)  Rooting depth extending**

The target function $S_m$ is defined as follows to calculate the rooting depth extension via optimisation:

$$S_m = \sum_{i=1}^{m}(thk_i \cdot fr_i \cdot REW_i) \tag{19}$$

When $S_m$ reaches its maximum, the optimum distribution of new roots over the soil profile is obtained. If $S_m$ reaches its maximum when $m$ is equal to $n$, the length of the coarse root remains unchanged; when $m$ is equal to $n + 1$, the coarse root penetrates the next soil layer.






## 2.2 Data

### 2.2.1 Site description

The study site is located in the Provincial Nature Forest Reserve, Yeheshan (34°31.76N, 107°54.67E; 1,090 m elevation), in the southern part of the Loess Plateau in China (Fig 3). The climate is semi-humid with an annual average temperature of

11.3 °C and precipitation of 570 mm (from Yongshou station, see 2.2.2). It is hot in summer and cold in winter, and precipitation occurs predominantly from May to October and varies significantly interannually. The artificially afforested black locust (*Robinia pseudoacacia* L.) dominates the vegetation species with an average height of 10 m (planted since 2000) and a density of 2,450 trees/ha$^2$ (Ma et al., 2017). The experimental plots were situated within a black locust forestland on an average slope of 8°. Instruments for the microclimate and soil water observations were installed. The thickness of the loess

soil is estimated to be more than 50 m, and the buried depth of groundwater is beyond that depth (Liu et al., 2010).

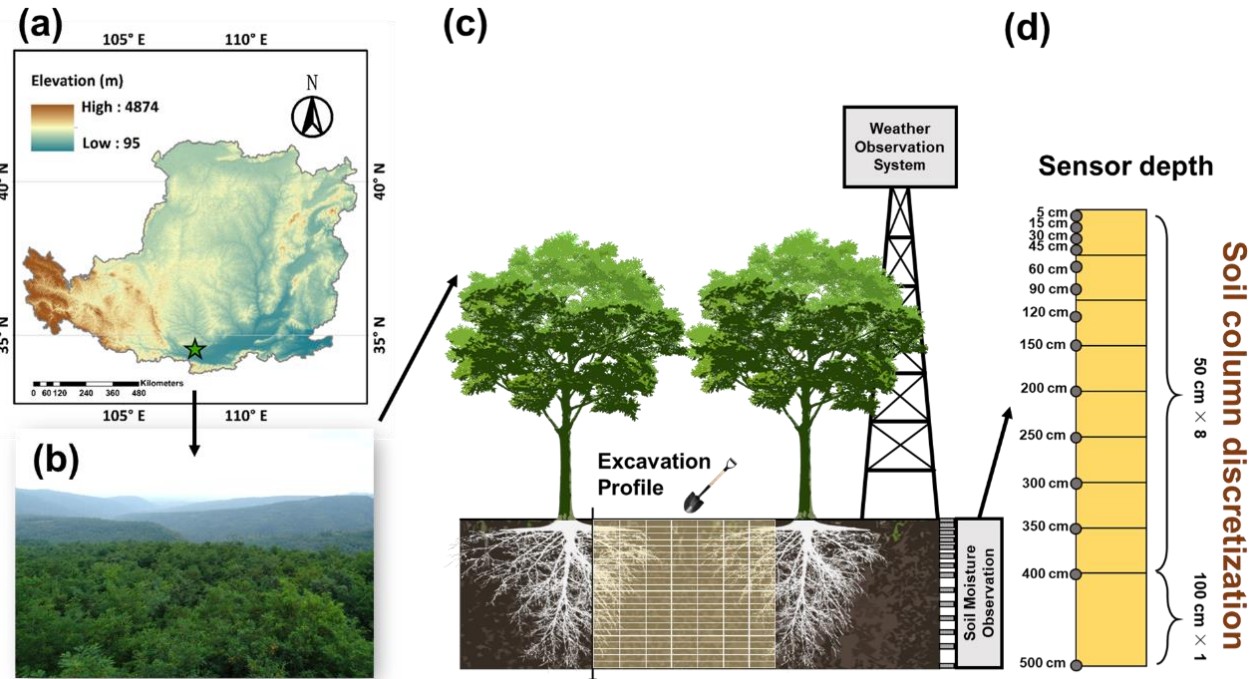

**Figure 3: (a) Location of the study site in the Loess Plateau, China; (b) top view from the black locust plantation canopy in July, 2018; (c) meteorological observation tower, soil moisture observation, and excavation root profile; (d) installation of the soil moisture sensors.**

### 2.2.2 Meteorology

A meteorological observation system was established on a flux tower in 2014. The tower was 16 m above the canopy (Fig 3). This system consists of sensors for temperature and humidity, HMP155A probes (Campbell Scientific, Inc., Logan, UT,





USA); for wind speed, the CSAT3 three-dimensional ultrasonic anemometer (Campbell Scientific, Inc., Logan, UT, USA); and for solar radiation, the CNR-4 net radiometer (Campbell Scientific, Inc., Logan, UT, USA). A T-200B precipitation

gauge (Geonor Inc., Oslo, Norway) was installed near the forest opening to measure throughfall. A CR3000 data logger (Campbell Scientific, Inc., Logan, UT, USA) was used to collect data from the sensors at 10-min intervals.

In addition, daily meteorological data from 1971 to 2020 for the National Metrological Station in Yongshou County, 26 km away from the field experiment site, were downloaded from the China Meteorological Data Service Centre (http://data.cma.cn) for the long-term simulation. The data series include daily precipitation, mean air temperature,

maximum air temperature, minimum air temperature, relative humidity, wind speed, and sunshine hours.

It is believed that the data series spanning 50 years covers the inter-annual variations in climatic factors, especially the alternating wet and dry periods (Fig 4). The annual average precipitation is 570 mm, with a standard deviation (STD) of 122 mm.

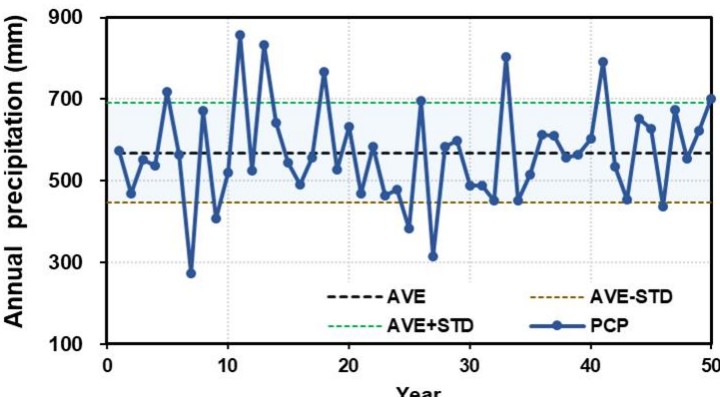

**Figure 4: Time series of annual precipitation (mm). Dashed coloured lines represent the average values (AVE) and plus or minus one standard deviation (STD).**

### 2.2.3 Soil and water

The soil properties were investigated by sampling over a profile 5 m below the ground at the study site. The dry bulk density ($\rho_b$, g cm$^{-3}$) was obtained by drying volumetric soil samples (100 cm$^3$) at 105 °C for 48 h, and the soil particle size

distribution and organic matter content were measured in the laboratory. Silt loam dominated the profile, with moderate variations among the soil layers. On average, the silt loam consisted of 5.8% sand, 73.4% silt, and 20.9% clay (Ma et al., 2017).

The saturated hydraulic conductivity ($K_s$, cm h$^{-1}$) was measured in the undisturbed soil samples using a constant-head method (Ramos et al., 2017). The field capacity ($\theta_{fc}$) and wilting point ($\theta_{wp}$) were derived from the power function

(Campbell, 1974; Clapp and Hornberger, 1978), corresponding to the soil potentials of −33 kPa and −1,500 kPa, respectively.





Volumetric soil moisture sensors were installed in 14 layers within the depth of 500 cm, at 5, 15, 30, 45, 60, 90, 120, 150, 200, 250, 300, 350, 400, and 500 cm, respectively (Fig 3). The sensors, CS-655 Soil Water Content Reflectometers (Campbell Scientific, Inc., Logan, UT, USA), have been in operation since June 2014. A CR1000 data logger (Campbell Scientific, Inc., Logan, UT, USA) records the data every 10 min.

The soil properties below 5 m were adopted from previous studies (Li et al., 2008; Jia et al., 2017; Wu et al., 2021) on black locust plantations in the Loess Plateau.

The soil desiccation index (SDI) was used to evaluate the degree of soil desiccation for the comparison between the observation and simulation results, calculated as:

$$SDI = \frac{\theta - \theta_{wp}}{\theta_{sfc} - \theta_{wp}} \tag{20}$$

where $\theta_{sfc}$ is the soil water content at a stable field capacity. In practice, soil water content at 60% of field capacity ($\theta_{fc}$) can be assumed to be the stable field capacity of loess in the Loess Plateau (Wang et al., 2011). Soil layers with SDI < 1 were regarded as the drying soil layers.

### 2.2.4 Plant

The leaf area index (LAI, $m^2\ m^{-2}$) of the black locust canopy was measured using an optical method (Jonckheere et al., 2004),
biweekly or triweekly during the growing seasons of 2014, 2015, and 2016. An 8 mm fisheye lens (Sigma F3.5 EX DG Circular Fisheye-Sigma Corporation) mounted on a Canon EOS 5D digital SLR camera (http://www.canon.com) took hemispherical photographs of the canopy on cloudy days. The photographs were analysed using the CAN_EYE software to derive the LAI values (Demarez et al., 2008). Within each plot, photographs were taken at five different positions each time. The LAI value for the plot is the average of the five positions.

The measured LAI values were compared to those of the Moderate Resolution Imaging Spectroradiometer (MODIS) product MYD15A2H, which has a spatial resolution of 500 m and a temporal resolution of 8 d. Furthermore, the 8-d MODIS LAI series were downscaled to a daily series using the Savitzky–Golay filtering technique (Tie et al., 2017).

The root profiles were investigated in August 2015. A 150-cm wide trench was excavated 500 cm below the ground, which was located between two neighbouring rows, perpendicular to the row direction (Fig 3). During the excavation, soil
samples, 20 cm in horizontal dimensions and 40 cm in vertical thickness, were taken along the trench and over the profile. It is assumed that the roots develop homogeneously along the horizontal row but unevenly along the trench. Finally, seven samples were collected from each soil layer across the trench. The soil samples were rinsed, and the weight and length of fine roots (less than 2 mm in diameter) were measured for each sample.

To estimate the parameters of the black locust root system, a pot experiment was carried out in 2016 (planted in April
and sampled in October), the details of which are given in **Fig S1** in the **Supplementary File**.



In addition to the measurements mentioned above, data on black locust roots, including the density profile and rooting depth were compiled from the published literature (see **Supplementary File**).

## 2.3 Model setup

### 2.3.1 Initial and boundary conditions for the Richards' equation

In solving Richards' equation numerically, the vertical domain extends from the surface to 20 m below the ground. The domain was discretized into adjoining layers with a thickness of 5 cm each.

The upper boundary condition was set as the flux of the rainfall rate with the canopy interception removed or the soil evaporation rate. The lower boundary was set to a constant soil water content at the field capacity.

During the calibration and validation stages, the initial soil water profile was determined using the measurements. The 300 initial soil water profile was set at the field capacity when the model was applied to the long-term simulation.

### 2.3.2 Numerical simulations

The soil hydraulic parameters, saturated hydraulic conductivity $K_s$ and constant $B$ in the soil water retention curve, were initialised by the measured values. They were further tuned to match the simulated soil water content to the observations.

The vegetation growth parameters were adapted from Zhang et al. (2015), who simulated the black locust growth in the 305 Loess Plateau using the Biome-BGC model. Other vegetation parameters were obtained from the SWAT model (Neitsch et al., 2011; Sun et al., 2011).

All the relevant parameters are summarized in **Table S1**.

### 2.3.3 Numerical simulations

(1) The model calibration and validation were performed for the observation period (2014–2018). The model was 310 calibrated from 1 June 2014 to 31 December 2016 and validated through 1 January 2017 to 31 December 2018. The measured LAI, soil water content, and root profiles were used for the evaluation. The rooting depth was assumed to be 5 m below the ground for the three approaches, considering the relatively short period of the field experiment.

(2) A long-term simulation was performed to explore the forest root–soil water interactions over a period of 50 years, with the aim of investigating the drying soil layer evolution over the long term under inter-annual variations of precipitation. 315 The long-term simulation adopted the data at Yongshou Station from 1971 to 2020 without any sense of a specific historical period. In the long-term simulation, a value of 5 m was set for the rooting depth of the static approaches; an initial value of 50 cm was set for the dynamic rooting depth approach. Plants start to grow at the beginning of the simulation.



### 2.3.4 Evaluation indices

Statistical indices, the coefficient of determination ($R^2$), Nash-Sutcliffe efficiency ($NSE$), and percent bias ($PBIAS$), were used to evaluate model performance and are given as follows, respectively:

$$R^2 = \frac{\sum_{i=1}^{n}(S_i - \bar{S})(O_i - \bar{O})}{\sqrt{\sum_{i=1}^{n}(S_i - \bar{S})\sum_{i=1}^{n}(O_i - \bar{O})}} \tag{21}$$

$$NSE = 1 - \frac{\sum_{i=1}^{n}(O_i - S_i)^2}{\sum_{i=1}^{n}(O_i - \bar{O})^2} \tag{22}$$

$$PBIAS = \frac{\sum_{i=1}^{n}(S_i - O_i)}{\sum_{i=1}^{n}O_i} \times 100\% \tag{23}$$

where $S_i$ is the simulated value at time step $i$, $\bar{S}$ is the mean of the simulated value, $O_i$ is the observed value at time step $i$, $\bar{O}$ is the mean of the observed value, and $n$ is the number of time steps. $R^2$ and $NSE$ were dimensionless. The dimension of the $PBIAS$ was %.

    $R^2$ describes the proportion of the variance in the measured data explained by the model, ranging from 0 to 1, with higher values indicating less error variance. $NSE$ indicates the consistency between the plot of the observed versus simulated data

and the 1:1 line, ranging between $-\infty$ and 1.0 (1 inclusive) and optimised at the value of 1. $PBIAS$ measures the average tendency of the simulated data to be larger or smaller than the observation, and a lower value imply a more accurate model simulation, with optimisation at 0.0. Positive values indicate a model underestimation bias, and negative values indicate a model overestimation bias. Moriasi et al. (2007) proposed a widely used rating system, which judged the modelling performance as "very good", "good", "satisfactory", or "unsatisfactory" using $PBIAS < \pm10\%$, $\pm10\% \leq PBIAS < \pm15\%$, $\pm15\%$

$\leq PBIAS < \pm25\%$, or $PBIAS \geq \pm25\%$, respectively; or by $0.75 < NSE \leq 1.0$, $0.65 < NSE \leq 0.75$, $0, 0.50 < NSE \leq 0.65$, or $NSE \leq 0.50$, respectively.

## 3 Results

### 3.1 Model calibration and validation

    The model parameters, maximum LAI ($\textbf{\textit{LAI}}_{max}$), saturated hydraulic conductivities ($\textbf{\textit{K}}_s$), and exponent of the soil-water

characteristic curve ($\textbf{\textit{B}}$), were calibrated and validated using the field measurements. The initial value of the $\textbf{\textit{LAI}}_{max}$ was assigned a default of 5.0 in the SWAT model; the initial values for $\textbf{\textit{K}}_s$ and $\textbf{\textit{B}}$ were based on measurements in the laboratory of the samples taken from the field site. The soil hydraulic parameters varied for the different soil layers. The calibration was performed manually using the observed LAI and soil water content as the target. The performance was evaluated using the indices mentioned in the previous section, and the calibrated values are listed in **Table S1**.

The simulated LAI values were plotted against the field measurements for 2014–2016 for calibration and 2017–2018 for validation (Fig 5). Further, the MODIS-derived LAI values were used for evaluating the simulation over the entire period as well, especially for the validation period of 2017–2018, for which the field measurements were not available.

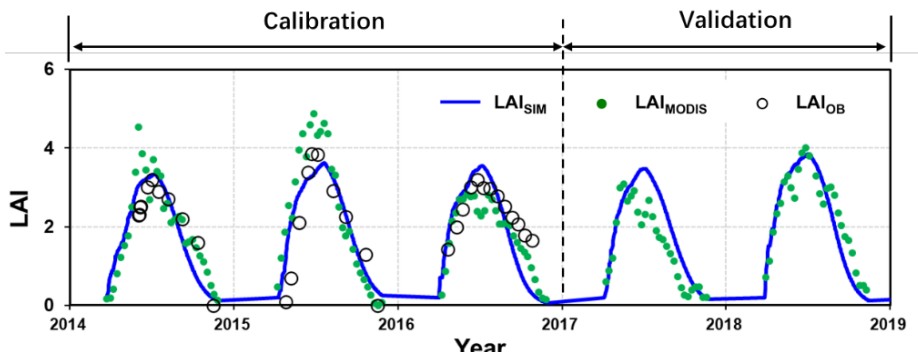

**Figure 5: Comparison of leaf area index (LAI) from simulation (LAI$_{SIM}$), MODIS-derived data (LAI$_{MODIS}$), and plot observation**
**(LAI$_{OB}$) for the calibration (2014–2016) and validation (2017–2018) periods.**

During the calibration period, the simulated LAI values fit the measurements with a classification of "very good"
(Moriasi et al., 2007), with a *NSE* of 0.60 and *PBIAS* of 5.2%. Validation with the MODIS-derived LAI indicated a "very
good" performance (Moriasi et al., 2007) with a *NSE* of 0.80 and *PBIAS* of 17.5%.

The MODIS-derived LAI exhibited remarkably similar seasonal patterns to the field measurements. Over- or under-
estimation was also noticed, for example, in 2014–2016 and 2017, respectively (Fig 5). Other studies have reported that
overestimation may occur especially for the LAI during the wet season when compared with the field experiments, for
example, by Yang (2006) and Naithani et al. (2013), or cross-evaluated against other remote sensing-based products, for
example, by Garrigues et al. (2013). The simulation demonstrated an overestimation of LAI when compared with the
MODIS-derived LAI in 2017. Although it is also argued that the MODIS-derived LAI may under-estimate the reality (Fang
et al., 2012), it is not sure which one (or both) is responsible for the discrepancy between the simulated and observed values
in 2017 due to non-availability of the field measurements. The point-pixel comparison issue might also be a reason for the
quantitative difference between the MODIS-derived and field-measured LAI. Nevertheless, the evaluation indices indicate
the acceptance of the *LAI$_{max}$* by the model performance in both the calibration and validation stages.

Comparisons between the simulated and measured soil water content (SWC) over the soil profile are provided in **Fig S2**
given in the **Supplementary File**. A simplified picture for comparing the averaged soil water content over the 5 m profile is
displayed in Fig 6, which illustrates the differences among the different root growth simulation approaches. The simulations
reproduced the change patterns of SWC over the seasons and between rainfall events exceptionally well in both the
calibration and validation stages. The dynamic fine root distribution approaches reproduced the variations in SWC
remarkably well. However, the SWC simulated using the static root distribution approach deviated significantly from the
measurements. The results of this approach will no longer be discussed in the forthcoming sections.





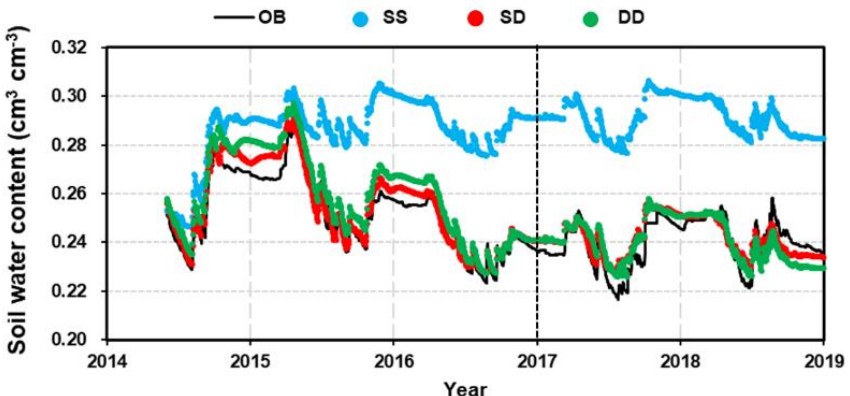

**Figure 6: Comparison of the 5 m profile averaged soil water content (SWC) from observation (OB) and three root simulation approaches during the calibration (2014–2016) and validation (2017–2018) period. Notes: SS, Static rooting depth and Static fine root distribution; SD, Static rooting depth and Dynamic fine root distribution; DD, Dynamic rooting depth and Dynamic distribution of fine roots.**

The evaluation indices $R^2$ for the static and dynamic rooting depth approaches were 0.96, 0.96, *NSE* 0.91 and 0.71, and *PBIAS* 1.2% and 2.9%, respectively, at the calibration stage; $R^2$ 0.66 and 0.64, *NSE* 0.66 and 0.58, and *PBIAS* 7% and 0.4%, respectively, at the validation stage. The performances can be categorised as "good" or "very good" following the rating system by Moriasi et al. (2007).

The fine root density profiles produced during the calibration and validation stages were compared with the sampled values obtained in August 2015 (Fig 7). The measured fine root densities varied significantly among the seven sampled profiles, and the variations decreased with soil depth. The simulated fine root densities exhibited an even wider range of variations, which covered the growth seasons from 2014 to 2018. On average, the root distribution of the dynamic rooting depth approach was closer to the measurements than that of the static rooting depth approach.

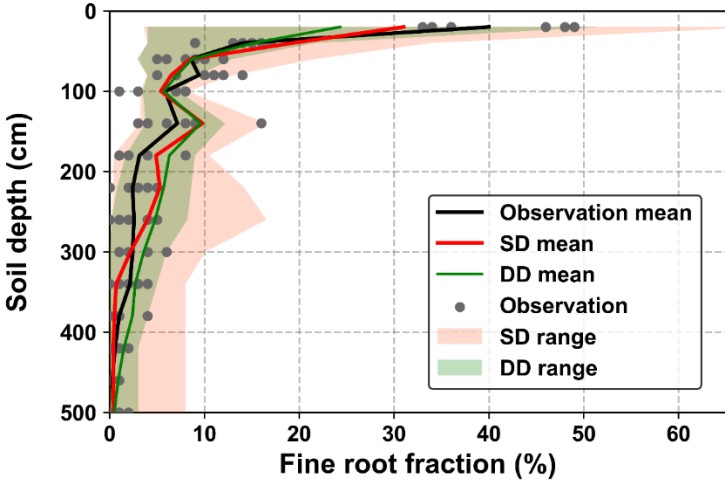





**Figure 7: Comparison of the 5 m profile averaged soil water content (SWC) from observation (OB) and three root simulation approaches during the calibration (2014–2016) and validation (2017–2018) period. Notes: SS, Static rooting depth and Static fine root distribution; SD, Static rooting depth and Dynamic fine root distribution; DD, Dynamic rooting depth and Dynamic distribution of fine roots.**

Variations in fine root distribution simulated by the static and dynamic rooting depth resulted from strategies for simulating the root–soil water interactions. In the static rooting depth approach, the growth of fine roots was purely determined by soil water availability and its distribution within the rooting domain, for example, 5.0 m in this study (see Eqs. (3) and (5) in Section 2.1.4). In contrast, in the dynamic rooting depth approach, growth of the fine roots may demand an increment of coarse roots that will cost more biomass and is finally determined by the optimisation of the cost-benefit

functions, as described by Eqs. (12) and (18). Thus, the dynamic root depth approach resulted in a narrower variation range than the static rooting depth approach. Averaged over the simulation period, the dynamic rooting depth approach achieved a more homogeneous fine root distribution profile than the static rooting depth approach, implying that the former approach utilised more soil water from the deeper soil layers. This point will be further addressed in the discussion regarding the drying soil layer evolution over the long term.

**3.2 Long-term simulation**

**3.2.1 Rooting depth and soil water**

Simulations forced by the long-term climatic data series revealed root development using the static and dynamic rooting depth approaches (Fig 8a). Instead of assigning a fixed root depth of 5.0 m in the static rooting depth approach, the dynamic rooting depth approach simulated the root depth extension, which is consistent with the data from the literature. It was found

that the rooting depth may be as deep as 11.0 m below the ground. The simulated rooting depth extension rate slowed as the stand age increased, which was also consistent with the observations in artificial forests (Christina et al., 2011; Wang et al., 2015) (see in **Fig S3**). Furthermore, the simulated root profiles by the two approaches within a 2.0 m soil depth were also evaluated against the data collected from the literature (Fig 8b&c). The simulated results varied within the range provided by the data in the literature. Regressions between the mean fine root fractions by the static and dynamic rooting depth

approaches and that of the literature data provided $R^2$ values of 0.34 and 0.64, respectively, indicating that the dynamic approach performed better than the static approach.

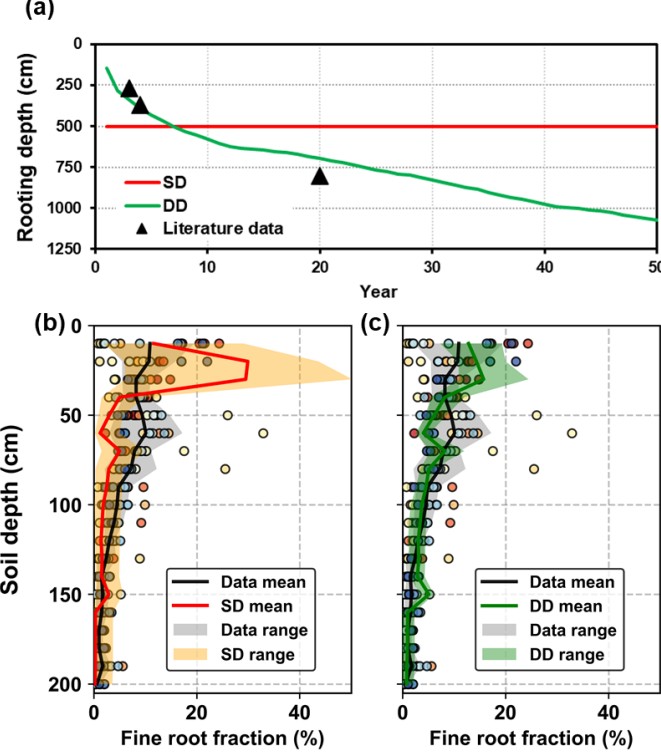

**Figure 8: Evaluation of the simulated root distribution against the literature data. (a) Rooting depth, (b) fine root distribution in top 2.0 m soil layer for the static rooting depth approach (SD), (c) fine root distribution in top 2.0 m soil layer for the dynamic rooting depth approach (DD). The shaded areas illustrate the range of mean ± standard deviation.**

The evolution of root growth and soil water over the long term is depicted in Fig 9. Visually, wetting and drying processes over the soil profiles were commonly found from simulations of both root growth modelling approaches. However, the difference in the simulated soil water distribution was also significant (Fig 9d).



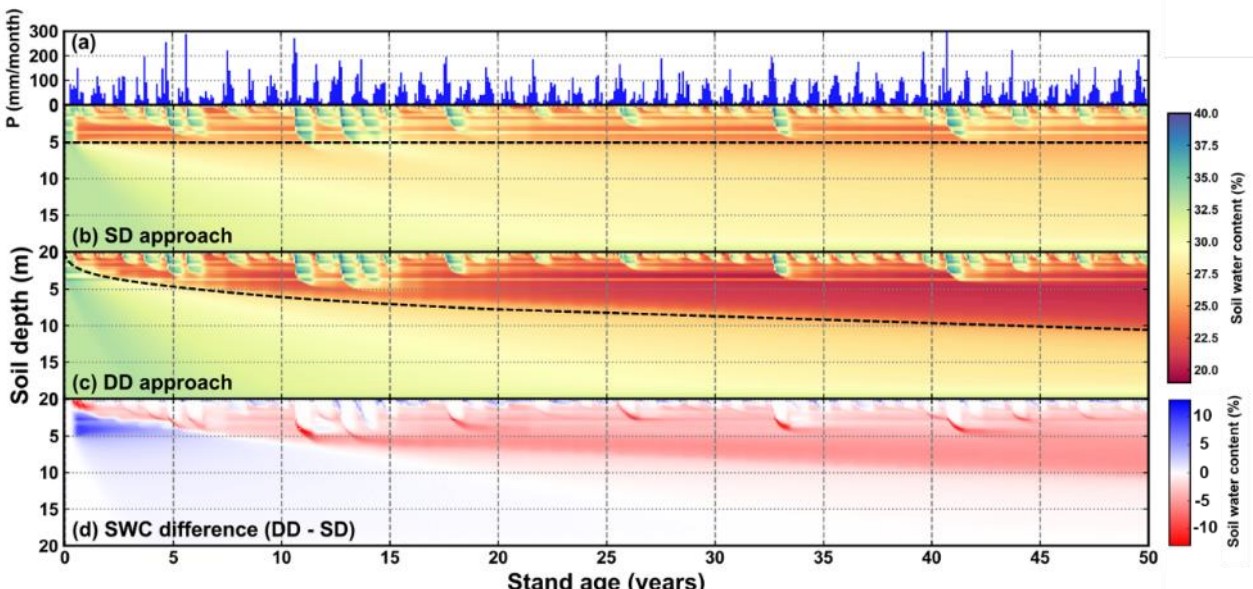

Figure 9: Time series of (a) monthly precipitation (mm month⁻¹), (b) soil water content (SWC) along the 20 m soil profile simulated by the static rooting depth (SD) approach, (c) soil water content along the 20 m soil profile simulated by the dynamic rooting depth (DD) approach, (d) the difference of SWC between the SD and DD approaches. The dashed black horizontal line in (b) and curve in (c) indicate the rooting depth.

These two approaches resulted in substantially different spatial distributions of roots and soil water, and thus a significant difference in root–water interactions. The soil water varied significantly with precipitation during the entire period. In most years, the maximum infiltration depth was less than 2.0 m. Meanwhile, precipitation could infiltrate down to 5 m in consecutive wet years, for example, the period from the 10th to 15th years. Notably, a time lag effect existed between the peak precipitation and the maximum infiltration depth; the peak precipitation occurred around August, while the maximum infiltration depth was reached around March in the following year.

### 3.2.2 Infiltration

Precipitation replenishes soil water through infiltration. The amount and depth of the infiltrated water impact the growth and water use of the root systems. The infiltration is associated with the a priori soil water content and its distribution over the profile, the amount and duration of an individual precipitation event, and the effects of the randomly sequential events. Establishing a relationship between the infiltration depth and precipitation on the basis of a single event is complicated and difficult to achieve. Instead, analyses were performed on an annual basis, that is, the maximum infiltration depth vs. the annual precipitation amount were regressed for the two root growth modelling approaches, as shown in Fig 10. During the simulation period of 50 years, the annual precipitation varied from 250 to 850 mm. It was found that the annual infiltration amounts by these two approaches were exceptionally close to each other. The dynamic rooting depth approach was 4.0%





lower than that of the static rooting depth approach, which is an insignificant difference, as discussed in a later section. The

maximum infiltration depth was positively correlated with the annual precipitation, which is not uncommon. The results also

indicated that the maximum infiltration depth may reach 6.0 m below the ground in the very wet years. Interestingly, the

regression lines of these two root growth modelling approaches crossed at approximately 500 mm of annual precipitation.

When annual precipitation was less than 500 mm, the infiltration reached deeper soil for the static rooting depth approach,

and vice versa when the annual precipitation was more than 500 mm.

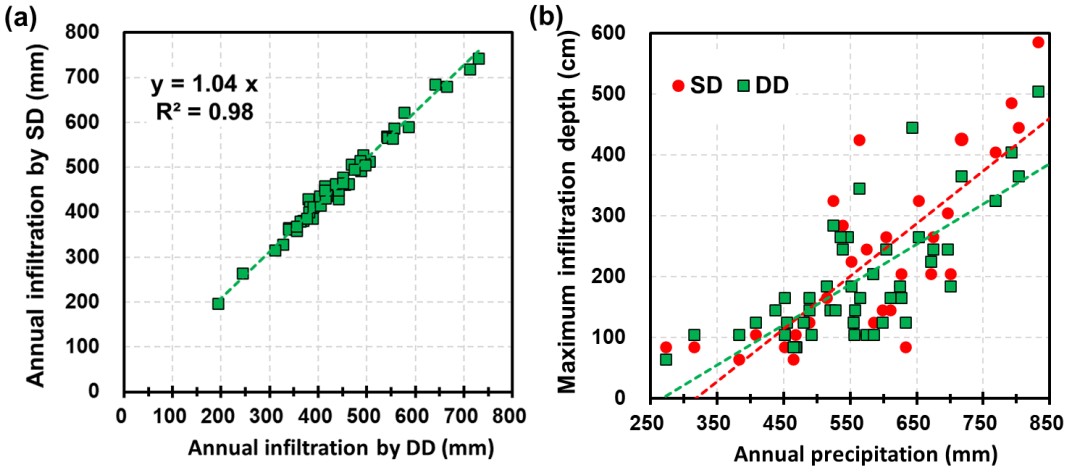


**Figure 10: Comparisons of (a) infiltration amounts and (b) maximum infiltration depths between the static rooting depth (SD) and dynamic rooting depth (DD) approaches**

### 3.2.3 Fine root distribution and water uptake

The fine root distributions simulated by these two approaches showed exceptionally similar patterns in the soil profile (Fig

11a). However, a quantitative difference between them was also noticeable. For the static rooting depth approach, roots grow

only within the soil layer with a present 5.0 m thickness. For the dynamic rooting depth approach, the coarse roots reach 11.0

m below the ground. Within the top 0.4 m soil layer, the fractions of fine roots for the static and dynamic rooting depth

approaches were 69% and 41%, respectively, and within the 0.4–5.0 m soil layer, 31% and 54%, respectively. For the

dynamic rooting depth approach, only 5% of the fine roots were located in the soil below 5.0 m.

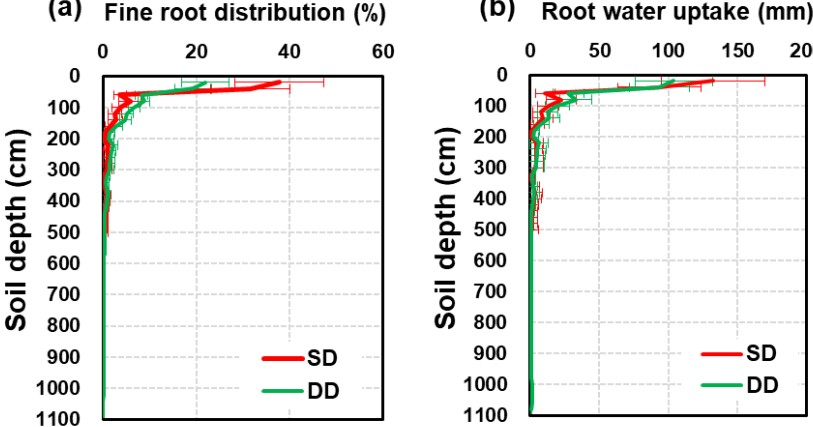


**Figure 11: Profile of (a) fine root density distributions (%) and (b) root water uptake distributions (mm) resulting from different approaches (SD and DD approach). The error bars represent the standard error (STD) of the mean.**

The distribution of root water uptake over the soil profile was similar to that of the fine root density (Fig 11b). The static rooting depth approach resulted in an annual water uptake of 338 mm (STD = 84 mm), while the dynamic rooting depth

approached 381 mm (STD = 84 mm). In the top 0.4 m soil layer, the root uptake was 226 mm and 198 mm for the static and dynamic rooting depth approaches, respectively, and 112 mm and 160 mm, respectively, in the 0.4–5.0 m layer. Below 5.0 m, the dynamic approach resulted in root uptake of 24 mm, which accounted for 6.2% of the uptake from the entire profile.

### 3.2.4 Evolution of drying soil layers

The long-term soil water evolution is illustrated in Fig 9. Vertically, the most active soil water zone was within the top 2.0 m,

statistically based on the simulation results. This was confirmed by root development and water uptake in this study (Fig 11) and by field observations in this region (Suo et al., 2018). In the top 2.0 m soil layer, infiltration events replenish and plant roots deplete the soil water alternately. The two root growth modelling approaches resulted in remarkably similar soil water enrichment and depletion patterns in the top 2.0 m layer, especially during the infiltration period (Fig. 9d). Below the top 2.0 m, soil water was continuously in a negative state, and the dynamic rooting depth approach resulted in more soil water

depletion than the static rooting depth approach.

Drying soil layers have been frequently reported in previous studies on the Loess Plateau (Wang et al., 2018b). The upper and lower boundaries of the drying soil layers were defined according to the soil desiccation index (Wang et al., 2011), as shown in Fig 12. The static rooting depth approach did not demonstrate the sustainable existence of the drying soil layers. For the dynamic rooting depth approach, the drying soil layers started to develop at a stand age of approximately 8 years and

became deeper and thicker sustainably with the increasing stand age. Its lower boundary extended gradually to deeper soil, while its upper boundary fluctuated significantly. The strong fluctuations were associated with the infiltration, while the





continuous extension of the lower boundary was due to sustained plant root uptake and rare recharge. With the stand aging, the soil water content in the drying soil layer decreased continuously, and the thickness increased gradually.

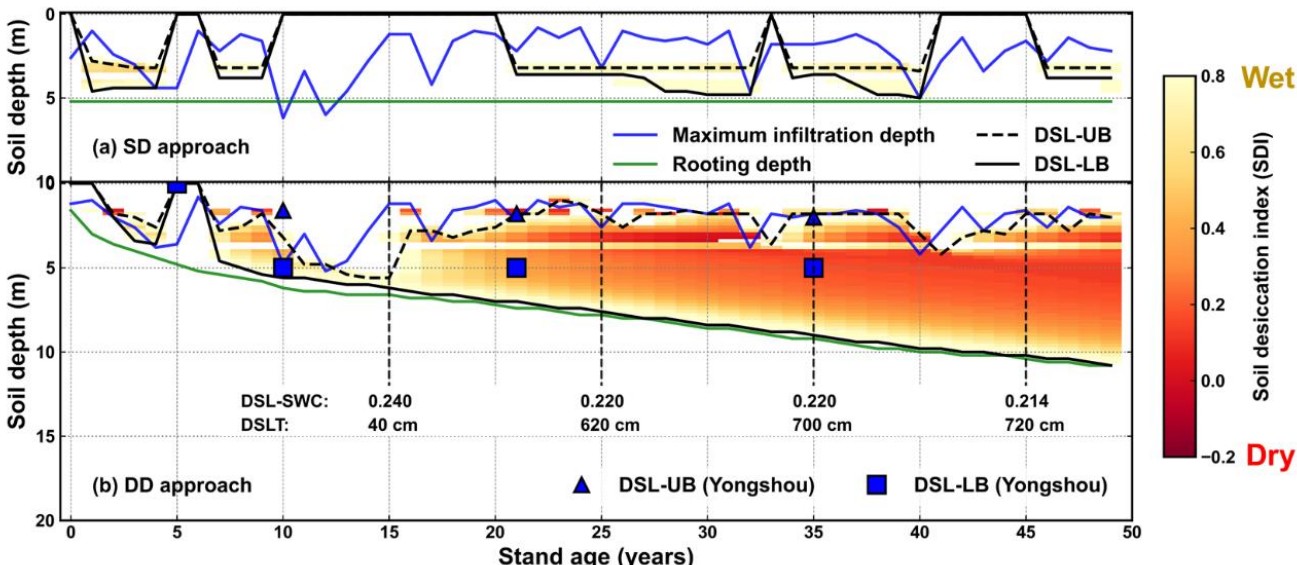

**Figure 12: Evolution of the drying soil layers with stand age simulated by (a) the static rooting depth (SD) and (b) the dynamic rooting depth (DD) approaches over a 50-year period at Yongshou, Loess Plateau. The triangles and squares are the observed upper and lower boundaries, respectively, adopted from Jia et al. (2017). Notes: DSL-UB: the upper boundary of drying soil layer; DSL-LB: the lower boundary of drying soil layer; DSL-SWC: mean soil water content within the drying soil layer; DSLT: the thickness of the drying soil layer.**

The simulated evolution of the drying soil layers was partially confirmed by field observations (Jia et al., 2017), as shown in Fig 12. The simulation produced a progressive course of the lower boundary, which was far deeper below the ground than the observations. The observations were limited to the top 5.0 m only. A deeper sampling might have resulted in a thicker drying soil. In Changwu, which is 80 km away from the study site in the Loess Plateau, it was reported that the thickness of the drying soil layer reached 7.0 m, with its lower boundary as deep as 8.0 m below the ground in the black

locust plantation (Li et al., 2008). It should also be noted that the sampling depth was limited at 8.0 m. Deep samplings have indicated that the drying soil layers reached the depth of 19.0 m in the Loess Plateau (Wu et al., 2021).



## 4 Discussion

### 4.1 Root–water interactions and drying soil layers

As indicated in the introduction section, this study attempted to develop a root growth model that can simulate coarse root extension dynamically instead of the static rooting depth approach currently adopted in most available models (Sivandran and Bras, 2013; Wang et al., 2016; Wang et al., 2018a; Drewniak, 2019; Niu et al., 2020). In formulating the dynamic rooting depth, the study proposed a cost–benefit algorithm based on ecophysiological principles (Chen et al., 2019). The evaluation of the simulated results against the field observations from this study and from the literature proved its

effectiveness for the Loess Plateau. Comparisons between the static and dynamic rooting depth approaches also determined that the former was incapable of reproducing the occurrence and evolution of the drying soil layers that have been widely reported in this region (Fig 12).

Notably, the development of the drying soil layers is predominantly due to water utilisation by the deep fine roots, which accounts for approximately only 5% of the total profile uptake (Fig. 11). Although minor compared with the total, it caused a

sustained negative soil water balance in the deep soil due to difficulties in receiving recharge, as described in the results section. The continuous development of the lower boundary of the drying soil layer implies that its recovery is critically difficult. This is because of the large thickness and vast storage capacity of loess soil (Huang and Shao, 2019). Plants tend to develop more fine roots in the topsoil and use more soil water due to lower costs but higher benefits, that is, a more profitable adaptation strategy when experiencing water stress. Exploration of water from wetter but deeper soil is also an

adaption strategy when it is more profitable, usually with more cost when coarse root growth requires additional biomass investment. This explains why the top 2.0 soil was the most active zone of water uptake in this study. Depletion of topsoil always vacates the storage for infiltration, making it difficult for the rainfall to replenish the deeper dried soil layer or groundwater (Turkeltaub et al., 2018).

The drying soil layer phenomenon has also been reported elsewhere, such as in the Amazonia forest (Jipp et al., 1998)

and southern Australia dryland (Robinson et al., 2006), due to artificial afforestation. The occurrence and development of the drying soil layer would have a significant impact on the plant–water relationship and the local hydrological cycling in the Loess Plateau, where the combination of deep loess soil and the semiarid and arid climate prevails (Zhao et al., 2019). This might also be an issue in other regions with similar soil and climate conditions (Shao et al., 2018).

### 4.2 Limitations of this work

Roots develop beneath the ground, which is widely accepted to be critically difficult to monitor; the deeper the soil depth, the harder it is to sample (Maeght et al., 2013; Warren et al., 2015; Fan et al., 2017). Observations of the rooting depths are rarely available for ready use, especially the knowledge of the maximum rooting depth for vegetation in a region. Occasionally, the limited data may provide a rough estimate, for example, Li et al. (2019) and Wu et al. (2021) reported





some maximum rooting depths of black locust and apple trees of approximately 25.0 m (Li et al., 2019; Wu et al., 2021).

Mathematical simulation is a beneficial compensation for field observations and can go far beyond its limitations, although its effectiveness relies on the actual data. The simulation can reproduce the dynamics of roots at very fine temporal and spatial resolutions, while in situ data are usually extremely rare. This study adopted in situ data at only a single moment, as shown in Fig 3. Fortunately, it was found that the measurements fell within the variation span of the simulations over some time periods, which is understandable from the statistical point of view. In situ data are believed to remain an issue in the

development and evaluation of root approaches in the long term (Pierret et al., 2016).

The simulation results indicated that the annual infiltration amount by the static rooting depth approach was 4.0% higher than that of the dynamic rooting depth approach (Fig 10). In the field, no surface runoff was found during the years of observation at the study site at Yeheshan. The minor difference in the infiltration amount between these two rooting depth approaches was due to canopy interception. The static rooting depth approach limits the roots to develop and use water

within the top 5.0 m soil layer, while the dynamic rooting depth approach permits the root to grow into deep wetter soil for uptake, which can release more soil water stress. The lower the soil water stress, the greater the leaf area and canopy storage. This small difference in the infiltration amount between these two approaches highlights the internal relationship between the above and belowground parts of the plants. The allocation of biomass between the above and belowground parts changes with climate, soil water, and nutrients over time and varies among vegetation types (Pooter et al., 2012; Qi et al., 2019).

These mechanisms were involved in the model of this study. However, this study adopted a fixed allocation factor, to focus on investigating the temporal and spatial changes in the rooting depth, fine root distribution, and their impacts on soil water evolution over the long term. Further attempts will be made to investigate the systematic behaviour of the plants on their adaptation strategies to obtain the maximum cost-benefit ratio by optimising the biomass allocation between the above and belowground parts (Trugman et al., 2019) and the spatial distribution of roots over the soil profile.

**5 Conclusion**

The study established a plant model and used it to investigate plant–water interactions in the Loess Plateau. This paper presents the results focusing on the root growth model and root–water interactions. The results indicate that incorporating the dynamic rooting depth into the currently available root growth model is a necessary step for an in-depth understanding of the occurrence and evolution of the widely reported drying soil layer phenomena in this region, which has also been reported

elsewhere.

The model provided a much more powerful tool to address the drying soil layer issue than in situ sampling techniques. It was found that the lower boundary of the drying soil layer extended sustainably due to the rare chances of replenishment by rainfall, while its upper boundary fluctuated strongly with the infiltration events. Continuous extension of the thickness of the drying soil layer and the difficulties in recovery may have strong implications for forest–water management in this region.




**Code and data availability.**

The meteorological datasets are available at http://data.cma.cn. The literature data datasets are available at in supplementary excel files. Further datasets and model code can be accessed upon request to the corresponding author.

**Author Contributions.**

Conceived and designed the research: YL and LS. Performed the experiments: HL, LS and XL. Collected and analysed the data: HL, XL, CM, TJ and HZ. Wrote and edited the paper: HL, YL and XW.

**Competing interests.**

The authors declare that they have no conflict of interest.

**Acknowledgement**

This work was supported by the National Key Research and Development Plan of China (NO. 2016YFC0501603) and the National Natural Science Foundation of China (NO. 41390461 and 41571130081). We also thank Dr. Jian Shi from Institute of Automation, Chinese Academy of Sciences, for his help for root image analysis. We would like to thank the anonymous reviewers of this paper for their helpful comments and suggestions.

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
