# Peer review of "Modelling the artificial forest (*Robinia pseudoacacia* L.) root-soil water interactions in the Loess Plateau, China"

_Hydrology and Earth System Sciences, 2021_

## Author Comment (AC1)

**Anonymous Reviewer #1**

Reviewer's comments are typed in **black** color, whereas the responses are typed in **blue** color.

**General comments:**

This article develops a root growth model which adjusts root distribution and rooting depth in the root water uptake model based on the cost-benefit theory, and the model verified by observational data is used to simulate the root depth and distribution from 1971 to 2020 to analyze and study the drying soil layers (DSLs), but this article is not thorough enough in some respects. It is of great practical significance for artificial afforestation to analyze the changes of the root system over water-scarce areas such as the Loess Plateau, and the regional analysis chart formed in the article has certain significance for various arid and semi-arid areas to carry out the regional ecological restoration.

As one of the important issues that this research focuses on, although DSLs have been extensively reported in artificial forest land, the issue should be introduced with a certain background in the introduction part.

Thanks. We will expand the introduction to the drying soil layer in the revised manuscript. See more details in the response of Line 503-513.

**Specific comments:**

Line 46: "complicated morphological distribution" should be "a complicated morphological distribution".

We will make the correction in the revised version.

Line 331: "imply" should be "implies". We will make the correction in the revised version.

Line 352 and 353: "a NSE" should be "an NSE". We will make the correction in the revised version. Line 361: "non-availability" should be "the non-availability". We will make the correction in the revised version.

Line 462: "the dynamic approach resulted in root uptake of 24 mm" should be "the dynamic approach resulted in a root uptake of 24 mm".

We will make the correction in the revised version.

Line 500-502: "Comparisons between the static and dynamic rooting depth approaches also determined that the former was incapable of reproducing the occurrence and evolution of the drying soil layers that have been widely reported in this region (Fig 12)." How this conclusion was obtained needs a more detailed and in-depth explanation.

Your suggestion is appreciated. The observations indicate that the occurrence, the upper and lower boundary, the soil water status within the drying soil layer change with time. The static rooting depth approach pre-sets a fixed root depth over the simulation period, which does not capture the hydraulic traits of roots that may develop to use water from the wetter zone beneath the pre-set root depth. We will add the explanation in the revised manuscript.

Line 509-511: "Exploration of water from wetter but deeper soil is also an adaption strategy when it is more profitable, usually with more cost when coarse root growth requires additional biomass investment." please provide evidence or reference for how this conclusion was obtained.

Appreciated. These understandings come truly from literature. We will cite some of these studies in the revised version, e.g., Pierret et al. (2016) and Germon et al. (2020). **References:**

Pierret, A., Maeght, J.-L., Clément, C., Montoroi, J.-P., Hartmann, C., and Gonkhamdee, S.: Understanding deep roots and their functions in ecosystems: an advocacy for more unconventional research, Ann. Bot., 118(4), 621-635, doi:10.1093/aob/mcw130, 2016. Germon, A., Laclau, J.P., Robin, A. and Jourdan, C.: Tamm Review: Deep fine roots in forest ecosystems: Why dig deeper? Forest Ecol. Manag., 466, 118135, doi:10.1016/j.foreco.2020.118135, 2020.

Line 503-513: "Notably, the development of the drying soil layers is predominantly due to water utilisation by the deep fine roots, which accounts for approximately only 5% of the total profile uptake (Fig. 11). Although minor compared with the total, it caused a 505 sustained negative soil water balance in the deep soil due to difficulties in receiving recharge, as described in the results section. The continuous development of the lower boundary of the drying soil layer implies that its recovery is critically difficult. This is because of the large thickness and vast storage capacity of loess soil (Huang and Shao,

2019). Plants tend to develop more fine roots in the topsoil and use more soil water due to lower costs but higher benefits, that is, a more profitable adaptation strategy when experiencing water stress. Exploration of water from wetter but deeper soil is also an adaption strategy when it is more profitable, usually with more cost when coarse root growth requires additional biomass investment. This explains why the top 2.0 soil was the most active zone of water uptake in this study. Depletion of topsoil always vacates the storage for infiltration, making it difficult for the rainfall to replenish the deeper dried soil layer or groundwater (Turkeltaub et al., 2018)." Pleas e supplement the significance of this research from a practical perspective in combination with the actual vegetation restoration situation on the Loess Plateau.

We appreciate this suggestion very much. As a matter of fact, Huang and Shao (2019) reviewed the studies on soil water in the Loess Plateau of northwest China. In this paper, the research progresses in the drying soil layer of the artificial forestation and their practical significance have been discussed in depth. In our manuscript, we focus on discussing mechanisms of the occurrence and evolution of the drying soil layer on basis of the mathematical simulation. In the revised version, we will try to enhance the discussion about the practical significance of this study by referring to the earlier review work by Huang and Shao (2019).

**References:**

Huang, L., & Shao, M. (2019). Advances and perspectives on soil water research in China'sLoessPlateau.Earth-ScienceReviews,199,102962.https://doi.org/10.1016/j.earscirev.2019.102962.

Figure 8: Since the circles on Figures 8b and 8c represent observation values, please explain what their different colors mean in the caption.

Appreciated. The explanations of the different colors will be added in the caption of Figure 8.

Figure 10b: The DD symbol has a black edge but the SD symbol does not. Please unify the style.

Appreciated. The plotting style of Figure 10b will be unified.

---

## Author Comment (AC2)

**Anonymous Reviewer #2**

*Reviewer's comments are typed in **black** color, whereas the responses are typed in blue color.*

**General comments:**

Based on the in situ observations, the manuscript proposes a root growth model that simulates both the dynamic rooting depth and fine root distribution. Subsequently, the model was used to simulate the forest-soil water relationships, including soil water availability and the temporal–spatial dynamics distribution of the dynamic rooting depth and fine root distribution in the Loess Plateau (LP) of China. Further, a long-term simulation was performed to address the drying soil layers issues in the region. The results show that incorporating the dynamic rooting depth into the currently available root growth models is necessary for accurately reproducing the drying soil processes. The manuscript is well-written and innovative. The proposed provides a much needed and powerful tool to address the drying soil layer issue than in situ sampling techniques. The findings on the thickness of the drying soil layer and the difficulties in recovery offer insight and strong implications for forest–water management in this region. The manuscript is of interest to the readers of the journal as well the wider ecohydrology community. I only have the following minor suggestions for the authors to consider:

Thanks for the valuable comments and suggestions. We revised the manuscripts following the comments and suggestions. The point-to-point explanation to the revisions are as follows.

**Specific comments:**

Yes, it is true how the black locust roots and soil water interact has not been addressed in previous modelling studies in the Loess Plateau of China, a brief and precise of similar findings/studies for other tree species in other regions/countries would help readers understand the current research gap and strengthen the innovative nature of the manuscript.

The suggestion is crucially important.

We accessed the latest publication that is published on Biogeosciences on 12 July 2021,

which is closely related to our work.

Sakschewski et al. (2021) reviewed the root growth approaches in the current Earth System Models and concluded that "none of those studies have acknowledged resource investment, timing and physical constraints of tree rooting depth within a competitive environment". To deal with this issue, they proposed a variable rooting depth approach for the LPJ model. Their results indicate that "variable tree rooting strategies are key for modelling the distribution, productivity and evapotranspiration of tropical evergreen forests". In this work, the maximum rooting depth is related to the tree height by a logistic growth function (Sakschewski et al., A5) and the vertical distribution of the fine roots follows a shape function (Sakschewski et al., A2).

In our work, root growth and soil water are coupled. Growth of the coarse and fine roots is determined by an optimization function that takes account of the ratio of water uptake and the biomass allocation between the coarse and fine roots.

The variable root approach by Sakschewshi et al. (2021) deals with the trade-off of biomass allocation between the above- and below-ground parts. Our work of dealing the similar issues is still on-going. It is hoped that better understandings of vegetation – water interactions in the semi-humid and semi-arid Loess Plateau can be achieved soon.

We incorporated the latest information into the revised manuscript.

**References:**

Sakschewski, B., von Bloh, W., Drüke, M., Sörensson, A. A., Ruscica, R., Langerwisch, F., Billing, M., Bereswill, S., Hirota, M., Oliveira, R. S., Heinke, J., & Thonicke, K. (2021). Variable tree rooting strategies are key for modelling the distribution, productivity and evapotranspiration of tropical evergreen forests. Biogeosciences, 18(13), 4091–4116. https://doi.org/10.5194/bg-18-4091-2021

P7 Eq. 6 defines the relationship between the coarse and fine roots. An addition of the definition of coarse and fine roots in the introduction would also clarify the potential confusion about the distinction of the two.

The definition of coarse (>2 mm diameter) and fine roots (<2 mm in diameter) in the classic approach is used in this study, and we will try to clarify this in introduction part of the revised version.

**References:**

Smithwick, E. A. H., Lucash, M. S., McCormack, M. L., & Sivandran, G. (2014). Improving the representation of roots in terrestrial models. Ecological Modelling, 291, 193–204. https://doi.org/10.1016/j.ecolmodel.2014.07.023

Jackson, R. B., Mooney, H. A., & Schulze, E.-D. (1997). A global budget for fine root biomass, surface area, and nutrient contents. Proceedings of the National Academy of Sciences, 94(14), 7362–7366. https://doi.org/10.1073/pnas.94.14.7362

P3, Line 79, "potentially beneficial…" should be "potentially benefit.."

We made the correction in the revised version.

---

## Author Response (AR1)

**Author's response on "Modelling the artificial forest (*Robinia pseudoacacia* L.) root-soil water interactions in the Loess Plateau, China" by Li et al.**

Reviewer comments are typed in **black** color, whereas the responses are typed in **blue** color.

**Anonymous Reviewer #1**

**General comments:**

This article develops a root growth model which adjusts root distribution and rooting depth in the root water uptake model based on the cost-benefit theory, and the model verified by observational data is used to simulate the root depth and distribution from 1971 to 2020 to analyze and study the drying soil layers (DSLs), but this article is not thorough enough in some respects. It is of great practical significance for artificial afforestation to analyze the changes of the root system over water-scarce areas such as the Loess Plateau, and the regional analysis chart formed in the article has certain significance for various arid and semi-arid areas to carry out the regional ecological restoration.

As one of the important issues that this research focuses on, although DSLs have been extensively reported in artificial forest land, the issue should be introduced with a certain background in the introduction part.

We are thankful for the valuable comments and suggestions.

First, we expanded the introduction to the drying soil layer in the revised manuscript. Please see lines 96-103. Then, we revised the manuscripts following the comments and suggestions. The point-to-point explanation to the revisions are as follows.

**Specific comments:**

Line 46: "complicated morphological distribution" should be "a complicated morphological distribution".

We made the correction in the revised version.

Line 331: "imply" should be "implies". We made the correction in the revised version.

Line 352 and 353: "a NSE" should be "an NSE".

We made the correction in the revised version.

Line 361: "non-availability" should be "the non-availability". We made the correction in the revised version.

Line 462: "the dynamic approach resulted in root uptake of 24 mm" should be "the dynamic approach resulted in a root uptake of 24 mm". We made the correction in the revised version.

Line 500-502: "Comparisons between the static and dynamic rooting depth approaches also determined that the former was incapable of reproducing the occurrence and evolution of the drying soil layers that have been widely reported in this region (Fig 12)." How this conclusion was obtained needs a more detailed and in-depth explanation.

Your suggestion is appreciated. The observations indicate that the occurrence, the upper and lower boundaries, and the soil water status within the drying soil layer change with time. The static rooting depth approach pre-sets a fixed root depth over the simulation period, which does not capture the hydraulic traits of roots that may advance to use water from the wetter zone beneath the pre-set root depth. We added the explanation in the revised manuscript. Please see lines 487-488 and 517-523.

Line 509-511: "Exploration of water from wetter but deeper soil is also an adaption strategy when it is more profitable, usually with more cost when coarse root growth requires additional biomass investment." please provide evidence or reference for how this conclusion was obtained.

Appreciated. These understanding comes from literatures. We cited some of these literatures in the revised version., e.g., Pierret et al. (2016) and Germon et al. (2020). Please see lines 530-532.

**References:**

Pierret, A., Maeght, J.-L., Clément, C., Montoroi, J.-P., Hartmann, C., and Gonkhamdee, S.: Understanding deep roots and their functions in ecosystems: an advocacy for more unconventional research, Ann. Bot., 118(4), 621-635, doi:10.1093/aob/mcw130, 2016. Germon, A., Laclau, J.P., Robin, A. and Jourdan, C.: Tamm Review: Deep fine roots in forest ecosystems: Why dig deeper? Forest Ecol. Manag., 466, 118135, doi:10.1016/j.foreco.2020.118135, 2020.

Line 503-513: "Notably, the development of the drying soil layers is predominantly due to water utilisation by the deep fine roots, which accounts for approximately only 5% of the total profile uptake (Fig. 11). Although minor compared with the total, it caused a 505 sustained negative soil water balance in the deep soil due to difficulties in receiving recharge, as described in the results section. The continuous development of the lower boundary of the drying soil layer implies that its recovery is critically difficult. This is because of the large thickness and vast storage capacity of loess soil (Huang and Shao, 2019). Plants tend to develop more fine roots in the topsoil and use more soil water due to

lower costs but higher benefits, that is, a more profitable adaptation strategy when experiencing water stress. Exploration of water from wetter but deeper soil is also an adaption strategy when it is more profitable, usually with more cost when coarse root growth requires additional biomass investment. This explains why the top 2.0 soil was the most active zone of water uptake in this study. Depletion of topsoil always vacates the storage for infiltration, making it difficult for the rainfall to replenish the deeper dried soil layer or groundwater (Turkeltaub et al., 2018)." Please supplement the significance of this research from a practical perspective in combination with the actual vegetation restoration situation on the Loess Plateau.

We appreciate this suggestion very much. As a matter of fact, Huang and Shao (2019) reviewed the studies on soil water in the Loess Plateau of northwest China. In this paper, the research progresses in the drying soil layer of the artificial forestation and their practical significance have been discussed in depth. In our manuscript, we focus on discussing mechanisms of the occurrence and evolution of the drying soil layer on basis of the mathematical simulation. In the revised version, we enhanced the discussion about the practical significance of this study by referring to the earlier review work by Huang and Shao (2019). Please see lines 539-545.

**References:**

Huang, L., & Shao, M. (2019). Advances and perspectives on soil water research in China'sLoessPlateau.Earth-ScienceReviews,199,102962.https://doi.org/10.1016/j.earscirev.2019.102962

Figure 8: Since the circles on Figures 8b and 8c represent observation values, please explain what their different colors mean in the caption.

We added the explanations to the colors in the caption of Figure 8.

Figure 10b: The DD symbol has a black edge but the SD symbol does not. Please unify the style.

We updated the Figure 10b.

Figure 1: Comparisons of (a) infiltration amounts and (b) maximum infiltration depths between the static rooting depth (SD) and dynamic rooting depth (DD) approaches

**Anonymous Reviewer #2**

**General comments:**

Based on the in situ observations, the manuscript proposes a root growth model that simulates both the dynamic rooting depth and fine root distribution. Subsequently, the model was used to simulate the forest-soil water relationships, including soil water availability and the temporal–spatial dynamics distribution of the dynamic rooting depth and fine root distribution in the Loess Plateau (LP) of China. Further, a long-term simulation was performed to address the drying soil layers issues in the region. The results show that incorporating the dynamic rooting depth into the currently available root growth models is necessary for accurately reproducing the drying soil processes. The manuscript is well-written and innovative. The proposed provides a much needed and powerful tool to address the drying soil layer and the difficulties in recovery offer insight and strong implications for forest–water management in this region. The manuscript is of interest to the readers of the journal as well the wider ecohydrology community. I only have the following minor suggestions for the authors to consider:

We thank that the reviewer thinks the paper is well-written and innovative. We appreciate all the useful suggestions that will improve the overall quality of our paper. We revised the manuscripts following the comments and suggestions. The point-to-point explanation to the revisions are as follows.

**Specific comments:**

Yes, it is true how the black locust roots and soil water interact has not been addressed in previous modelling studies in the Loess Plateau of China, a brief and precise of similar findings/studies for other tree species in other regions/countries would help readers understand the current research gap and strengthen the innovative nature of the manuscript.

Thanks for the crucial comment and suggestion.

Fortunately, we accessed the latest work of Sakschewski et al. that is published on Biogeosciences on 12 July 2021, which is closely related to our work.

Sakschewski et al. (2021) reviewed the root growth approaches in the current Earth system models and concluded that "none of those studies have acknowledged resource investment, timing and physical constraints of tree rooting depth within a competitive environment". To deal with this issue, they proposed a variable rooting depth approach for the LPJ model. Their results indicate that "variable tree rooting strategies are key for modelling the distribution, productivity and evapotranspiration of tropical evergreen forests".

In this work, it is assumed that the maximum rooting depth is related to the tree height by a logistic growth function (Sakschewski et al., A5) and the vertical distribution of the fine

roots follows a shape function (Sakschewski et al., A2).

In our work, the rooting depth and fine root distribution are constrained by the soil water distribution over the soil profile. The rooting depth and fine root distribution are finally determined by an optimization function that takes account of the ratio of water uptake and connections between the coarse and fine roots.

The variable root approach by Sakschewshi et al. (2021) deals with the trade-off of biomass allocation between the above- and below-ground parts. Our work of dealing the similar issues is still on-going. It is hoped that better understandings of vegetation – water interactions in the semi-humid and semi-arid Loess Plateau can be achieved soon.

We incorporated the latest information into the revised manuscript. Please see lines 68-75, 86-88, 103, and 571.

**References:**

Sakschewski, B., von Bloh, W., Drüke, M., Sörensson, A. A., Ruscica, R., Langerwisch, F., Billing, M., Bereswill, S., Hirota, M., Oliveira, R. S., Heinke, J., & Thonicke, K. (2021). Variable tree rooting strategies are key for modelling the distribution, productivity and evapotranspiration of tropical evergreen forests. Biogeosciences, 18(13), 4091–4116. https://doi.org/10.5194/bg-18-4091-2021

P7 Eq. 6 defines the relationship between the coarse and fine roots. An addition of the definition of coarse and fine roots in the introduction would also clarify the potential confusion about the distinction of the two.

The suggestion is appreciated. The definition of coarse (>2 mm diameter) and fine roots (<2 mm in diameter) in the classic approach is used in this study, and we clarified this in introduction part of the revised version. Please see lines 76-78.

**References:**

Smithwick, E. A. H., Lucash, M. S., McCormack, M. L., & Sivandran, G. (2014). Improving the representation of roots in terrestrial models. Ecological Modelling, 291, 193–204. https://doi.org/10.1016/j.ecolmodel.2014.07.023

Jackson, R. B., Mooney, H. A., & Schulze, E.-D. (1997). A global budget for fine root biomass, surface area, and nutrient contents. Proceedings of the National Academy of Sciences, 94(14), 7362–7366. https://doi.org/10.1073/pnas.94.14.7362

P3, Line 79, "potentially beneficial..." should be "potentially benefit.." We made the correction in the revised version.

---

## Editor Decision (ED1)

[revised manuscript text omitted]

**This seems to me not a conclusion section, it looks more like a summary. I suggest a longer summary.**

The study established a plant model and used it to investigate plant–water interactions in the Loess Plateau. This paper presents the results focusing on the root growth model and root–water interactions. The results indicate that incorporating the dynamic rooting depth into the currently available root growth model is a necessary step for an in-depth understanding of the occurrence and evolution of the widely reported drying soil layer phenomena in this region, which has also been reported elsewhere.

The model provided a much more powerful tool to address the drying soil layer issue than *in situ* sampling techniques. It was found that the lower boundary of the drying soil layer extended sustainably due to the rare chances of replenishment by rainfall, while its upper boundary fluctuated strongly with the infiltration events. Continuous extension of the thickness of the drying soil layer and the difficulties in recovery may have strong implications for forest–water management in this region.

**Can we compare the modeling method and observation method? It seems strange to make such a comparison.**

[revised manuscript text omitted]

---

## Author Response (AR2)

**Author's response on "Modelling the artificial forest (*Robinia pseudoacacia* L.) root-soil water interactions in the Loess Plateau, China" by Li et al.**

*Editor's comments are typed in **black** color, whereas the responses are typed in **blue** color.*

We are thankful for the valuable comments and suggestions. We revised the manuscripts following the comments and suggestions. The point-to-point explanation to the revisions are as follows.

1. Abstract:
a) The argument that incorporating dynamic soil depth is necessary for reproducing the drying soil process is an important point. It needs more explanation with concise results in the abstract.
b) The important role of dynamic root depth information of the drying soil layer is not *adequately* explained.
We refined the abstract according to your comments, please see lines 19-32.

2. Introduction: again, why root dynamics is important for the understanding of drying soil layer? It is not clear in the introduction part. It seems the two aspects are parallelly discussed and their connection is not clearly expressed.
We recomposed the introduction section following your comments, please see lines 88-100.

3. The lower boundary condition for 1D Richards' equation is field capacity. Any supporting evidence for such a configuration?
Field investigation indicates that the soil water content is relatively stable around the field capacity at depth of 20-100 m in Yangling, Loess Plateau (Qiao et al., 2018). We added this reference to the main text. Please see lines 308-310.
**References:**
Qiao, J., Y. Zhu, X. Jia, L. Huang, and M. Shao.: Factors that influence the vertical distribution of soil water content in the Critical Zone on the Loess Plateau, China, Vadose Zone J, 17, 170196, 10.2136/vzj2017.11.0196, 2018.

4. What's the reason that the static method cannot capture the soil water variation? Previous modeling work without root dynamics does not do such a bad job as shown in Figure 6. This needs more explanations.
Thanks for your comment. We made some modifications in Figure 6 in the revised version.

We updated the parameters of the SS (static root distribution and static rooting depth) method by the observed root distribution (Fig.A1) and updated Figure 6 in the main text. Along the soil profile, The SS method used less soil water from the deep layer of 100-400 cm than the SD and DD methods (Fig.A2, also **Fig.S3** in **Supplementary File**). The SS method reproduced the changing patterns of soil water over time, although the results of it shows much more deviation to the observations than the SD and DD methods (Figure 6 in the main text).

[Figure]

**Fig. A1 The fitted root density distribution over soil profile for the SS method**

[Figure]

**Figure A2: The simulated and observed average soil water content over the soil profile**

The SS method is widely used in different ecological or hydrological models. In the Loess Plateau, these models have been used to simulate soil water variations in field crops, shrubs, and forests (Table A1). Short-term model calibration and validation, ranging from months to five years, has usually shown acceptable performance, while long-term evaluation has rarely been undertaken. When used to address the long-term issue in this study, comparisons between the static and dynamic rooting depth approaches indicated that the former did not reproduce the occurrence and evolution of the drying soil layers due to its pre-set rooting depth (Figure 12 in the main text). We revised the discussion section, please see lines 513-524.

**Table A1 root models used in some publications the in the Loess Plateau**

| Literature | Hydrological or ecological Models | Vegetation types | Root distribution | Rooting depth (cm) | Simulation period |
|---|---|---|---|---|---|
| Zhang et al. (2015) | Modified Biome-BGC with 1D Darcy's equation | Forest | Exponential distribution | 100 | Calibration and validation: 2003-2006 Simulation: 1944–2007 |
| Tian et al. (2016) | WAVES | Forest | Exponential distribution | 100 | Calibration and validation: June 2011 to September 2011 Simulation: 1980–2010 |
| Li et al. (2019) | Hydrus-1D | Apple | Observed distribution obtained from local sampling | 620 | Calibration and validation: 2011-2013 Simulation: 1960-2013 |
| Bai et al. (2020) | Hydrus-1D | Crop, grass and shrub | Linear distribution or observed distribution collected from literature | 400 | Calibration and validation: 2004-2016 Simulation: 1970-2060 |

**Reference**

Zhang, Y., Huang, M., & Lian, J. (2015). Spatial distributions of optimal plant coverage for the dominant tree and shrub species along a precipitation gradient on the central Loess Plateau. Agricultural and Forest Meteorology, 206, 69–84. https://doi.org/10.1016/j.agrformet.2015.03.001

Tian, F., Feng, X., Zhang, L., Fu, B., Wang, S., Lv, Y., & Wang, P. (2016). Effects of revegetation on soil moisture under different precipitation gradients in the Loess Plateau, China. Hydrology Research, 48(5), 1378–1390. https://doi.org/10.2166/nh.2016.022

Li, B., Wang, Y., Hill, R. L., & Li, Z. (2019). Effects of apple orchards converted from farmlands on soil water balance in the deep loess deposits based on HYDRUS-1D model. Agriculture, Ecosystems & Environment, 285, 106645. https://doi.org/10.1016/j.agee.2019.106645

Bai, X., Jia, X., Jia, Y., Shao, M., & Hu, W. (2020). Modeling long-term soil water dynamics in response to land-use change in a semi-arid area. Journal of Hydrology, 585, 124824. https://doi.org/10.1016/j.jhydrol.2020.124824

5. Please double-check Figure 7 (the title is wrong?).

Thanks, we made the correction of the caption of Fig 7 in the revised version.

6. It is hard to tell the difference between SD and DD simulations compared to observation.

Yes, the differences between these two approaches are not remarkable over the ~4 years of evaluation period. However, significant difference appears when they are used to address the long-term issues, e.g., 50 years in this study. The main reason for that is the SD method presets a maximum rooting depth, e.g., 5 m in this study. However, the DD method let the rooting depth develops to use soil water available in deeper soil layer.

---

## Author Response (AR3)

**Author's response on "Modelling the artificial forest (*Robinia pseudoacacia* L.) root-soil water interactions in the Loess Plateau, China" by Li et al.**

*Editor's comments are typed in **black** color, whereas the responses are typed in **blue** color.*

I went through the revised manuscript and the authors' response. I am happy with the scientific content of the revision. However, based on my reading on abstract and conclusion parts, I still find significant improvements on the language aspects should be done before its final publication on HESS. Please check the attached annotated document for my comments. Please go through the whole manuscript and make a thorough revision. Thanks.

We are thankful for the valuable comments and suggestions. We revised the manuscripts following the comments and suggestions. The point-to-point explanation to the revisions are as follows.

1. Abstract: "dominate?" in line 18.
Thanks, we made the correction in the revised version. Please see lines 18.

2. Conclusion: This seems to me not a conclusion section, it looks more like a summary. I suggest a longer summary. Can we compare the modeling method and observation method? It seems strange to make such a comparison.
We recomposed the Conclusion section and converted it to Summary following your comments, please see lines 584-603.